

# DNA barcode-based survey of Trichoptera in the Crooked River reveals three new species records for British Columbia

Daniel J. Erasmus[1,*], Emily A. Yurkowski[1] and Dezene P.W. Huber[2,*]

[1] Department of Biochemistry and Molecular Biology, University of Northern British Columbia, Prince George, British Columbia, Canada

[2] Department of Ecosystem Science and Management Program, University of Northern British Columbia, Prince George, British Columbia, Canada

[*] These authors contributed equally to this work.

## ABSTRACT

Anthropogenic pressures on aquatic systems have placed a renewed focus on biodiversity of aquatic macroinvertebrates. By combining classical taxonomy and DNA barcoding we identified 39 species of caddisflies from the Crooked River, a unique and sensitive system in the southernmost arctic watershed in British Columbia. Our records include three species never before recorded in British Columbia: *Lepidostoma togatum* (Lepidostomatidae), *Ceraclea annulicornis* (Leptoceridae), and possibly *Cheumatopsyche harwoodi* (Hydropsychidae). Three other specimens may represent new occurrence records and a number of other records seem to be substantial observed geographic range expansions within British Columbia.

Corresponding authors
Daniel J. Erasmus, erasmus@unbc.ca, daniel.erasmus@unbc.ca
Dezene P.W. Huber, huber@unbc.ca

## INTRODUCTION

With accelerating anthropogenic climate change there is a renewed interest in assessing biodiversity in freshwater ecosystems (*Parmesan, 2006*). Freshwater ecosystems are especially under cumulative threats with increased demand for fresh water by industrial activities in riparian zones (*Meyer, Sale & Mulholland, 1999*). Assessing insect biodiversity is a challenging, but vital, activity in the face of these changes in order to understand aquatic food webs, ecosystem services, and for use in aquatic environmental monitoring (*Burgmer, Hillebrand & Pfenninger, 2007*; *Dobson & Frid, 2009*; *Cairns Jr & Pratt, 1993*).

Trichoptera taxonomy is primarily based on male adult morphology, which often requires experts for definitive identification. Taxonomy of the larvae is complicated and often problematic as it is not always possible to distinguish between species of the same genus (*Burington, 2011*; *Ruiter, Boyle & Zhou, 2013*). DNA barcoding and the use of sequence databases, combined with classical taxonomy, can help to speed up this process by allowing rapid surveys of novel regions (*Ruiter, Boyle & Zhou, 2013*; *DeSalle, Egan & Siddall, 2005*; *Jinbo, Kato & Motomi, 2011*; *Pauls et al., 2010*; *Zhou, Kjer & Morse, 2007*). The Barcode Of Life Database (BOLD) currently contains DNA barcodes for more than

260,000 species including ~4,555 Trichoptera species, and facilitates the identification of species based on subunit I of the cytochrome oxidase I (COI) DNA gene. In addition, recent comprehensive work on barcode-assisted Trichoptera taxonomy (*Zhou et al., 2009*; *Zhou et al., 2010a*; *Zhou et al., 2010b*; *Zhou et al., 2011*; *Zhou et al., 2016*) provides a solid foundation for biodiversity surveys of caddisflies in North America. Trichoptera, Ephemeroptera (mayflies), Plecoptera (stoneflies), and often aquatic Diptera (true flies) are used in well-developed protocols as indicators of aquatic ecosystem health (*Lenat & Barbour, 1994*). Due to their taxonomic richness, differential susceptibility to pollutants, and abundance in almost all water bodies worldwide, shifts in their numbers, relative ratios, or taxonomic diversity both temporally and/or geographically are used to observe stability and disturbance of ecosystems (*Houghton, 2004*; *Pond, 2012*). Monitoring work is best accomplished with good information on which species are present. Due to a lack of historical sampling in some areas, managers often must rely on regional (often province- or state-level) checklists that may or may not represent the taxonomic and functional diversity of smaller areas or specific sensitive systems. The Crooked River (Fig. 1) is the southernmost lotic system in British Columbia that ultimately drains into the Arctic Ocean. It flows north from Summit Lake (which is just on the north side of the continental divide) to McLeod Lake, connecting a series of lakes along the way. From there its water flows via other systems to eventually end up in the Williston Reservoir—a massive hydroelectric reservoir in the Rocky Mountain Trench that represents one of the largest anthropogenic landscape modifications on earth.

The Crooked River is named for all the oxbows due to its slow meandering flow (*McKay, 2000*). This river is also fed by underground springs, such as Livingston Springs in Crooked River Provincial Park. This well-known spring supplies the river with water year round and moderates annual temperature shifts. An extinct volcano (Teapot Mountain) is situated at its headwaters, and likely provides mineral nutrient inputs. As a *bona fide* spring creek, the Crooked River has a very flat gradient with swamp and marshland along much of its shoreline. During freshet the river floods these marshes bringing more nutrients into the system. These factors result in high productivity and a fairly stable year-round temperature which make the Crooked River unique compared to neighbouring systems. Nearby river systems are more typical of British Columbia—they are best described as oligotrophic freestone rivers that are highly susceptible to drastic changes in discharge from spring freshets and that show considerable annual temperature variation. The watershed has been logged for years resulting in a network of resource roads and bridges. A major highway and a rail line also run along much of its length, and are at times only a few meters from the river's main channel. However, even with its unique nature and high levels of anthropogenic impacts, our searches have revealed no recorded biodiversity surveys on the Crooked River.

Besides that, to our knowledge no comprehensive recent assessment has been done on Trichoptera in central or northern British Columbia. As the Crooked River is such a unique and nutrient-rich system we questioned whether it may provide habitat to species not yet reported for British Columbia. The aim of this study was to provide a comprehensive list

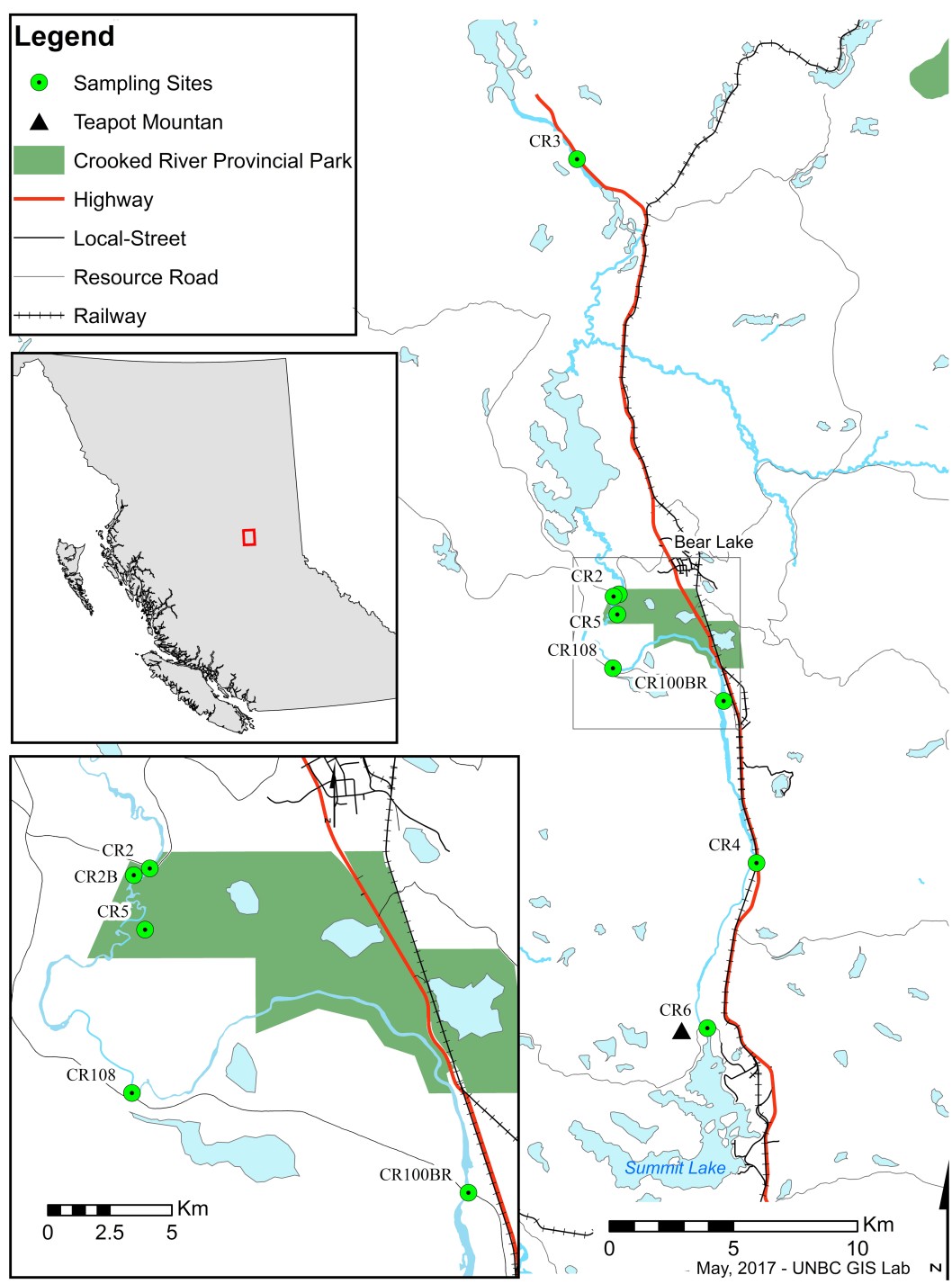

**Figure 1** **Map of sampling sites along the Crooked River, British Columbia.** CR2: 54.485265°N, −122.717974°W; CR2B: 54.484474°N, −122.721257°W; CR3: 54.642963°N, −122.743021°W; CR4: 54.387709°N, −122.633217°W; CR5: 54.477975°N, −122.719000°W; CR6: 54.328038°N, −122.669236°W; CR100BR: 54.446455°N, −122.653129°W; CR108: 54.458511°N, −122.721828°W.

of the Trichoptera biodiversity in a unique and vulnerable river as a baseline for future work and management.

## METHODS AND MATERIALS

We collected specimens on a biweekly basis from eight locations (CR2—54.484°N, −122.721°W, CR2B—54.484°N, −122.721°W, CR3—54.643°N, −122.743°W, CR4—54.388°N, −122.633°W, CR5—54.478°N, −122.719°W, CR6—54.328°N, −122.669°W, CR100BR—54.446°N, −122.653°W, CR108—54.458°N, −122.722°W) along the edge of the Crooked River, British Columbia between May and August 2014 using both hand and kick-net methods. This study focused mainly on larvae to ensure that we collected caddisflies from the Crooked River only and not from nearby water bodies. We completed collections under the British Columbia Ministry of Environment Park Use Permit #107171 where required. We preserved specimens in 80% ethanol upon collection. We identified all 2,204 caddisfly specimens that we collected to the lowest possible taxonomic ranking (genus or family) based on published morphological keys (*Wiggins, 1977*; *Clifford, 1991*; *Schmid, 1998*). We selected morpho-species and 214 specimens were subsequently sent to the Biodiversity Institute of Ontario (BIO) and its Barcode of Life Database (http://www.boldsystems.org) in Guelph, Ontario, to have their barcode region (COI) sequenced for further classification. We received back 185 useable sequences (>400 bp., <5 miscalls, no contamination detected). We vouchered all specimens sent for sequencing at the Centre for Biodiversity Genomics at the University of Guelph. Initial species identification was based on a 650 bp sequence in CO1 5′ region using the BOLD platform with MUSCLE sequence alignments and a Kimura-2-parameter distance model. The data for all collected specimens are available as dataset 10.5883/DS-CRTRI.

Neighbor joining analyses were performed on *Cheumatopsyche harwoodi*, *Lepidostoma togatum* and *Ceraclea annulicornis* specimens from the Crooked River compared to con- and heterospecific sequence data from the Barcode Of Life Database (BOLD). Evolutionary distances were computed using the Kimura 2-parameter method bootstrapped (5,000 replications) after a MUSCLE alignment and were visualized in MEGA6.0 (*Saitou & Nei, 1987*; *Felsenstein, 1985*; *Kimura, 1980*; *Tamura et al., 2013*). We cross-referenced the Crooked River Trichoptera species list that we obtained from analysis of our BOLD data using checklists, museums records and databases from the following: Canadian National Collection of Insects, Arachnids and Nematodes (http://www.canacoll.org/); Strickland Museum at the University of Alberta; Beaty Biodiversity Museum at the University of British Columbia; Electronic Atlas of the Wildlife of British Columbia (http://ibis.geog.ubc.ca/biodiversity/efauna/); Natureserve (http://www.natureserve.org/); Canadensys (http://www.canadensys.net/); Global Biodiversity Information Facility (http://www.gbif.org/); the Royal Ontario Museum; and the Royal British Columbia Museum (http://search-collections.royalbcmuseum.bc.ca/Entomology).

## RESULTS & DISCUSSION

We used morphological keys to identify all 2,204 collected specimens to family or genus, after which we used successful barcodes and database searches to deduce the species identities of 185 individuals based on previous database annotations. In total we detected 41 caddisfly species—found in 20 genera within 11 families—in the Crooked River system (Table 1). All barcode data are publicly available at BOLD (10.5883/DS-CRTRI). Thirty five of the 41 species we identified were assigned to known species via database matches using a 2% threshold for delineating species within Trichoptera, which is considered to be a reliable approach (*Zhou et al., 2009*). COI sequences of specimens from the Crooked River with DNA sequences matching 99.67% and 99.13% to *Lepidostoma cinereum* and *Neophylax rickeri* respectively, were assigned to the aforementioned species.

Among the 34 specimens identified to species with 100% database matches are *Cheumatopsyche harwoodi, Lepidostoma togatum* and *Ceraclea annulicornis,* all three are new species records for British Columbia.

There are currently six species within the genus *Cheumatopsyche* known from British Columbia: *C. analis, C. campyla, C. gracilis, C. oxa, C. pettiti* and *C. smithi* (http://ibis.geog.ubc.ca/biodiversity/efauna, *Cannings, 2007*). We found a larva of *Cheumatopsyche harwoodi* (synonym *C. enigma* Ross, Morse, & Gordon, 1971) at CR4 on May 16th 2014. Based on morphological keys we were only able to classify our specimen to genus level. This is not surprising as morphology-based taxonomy of *Cheumatopsyche* larvae is exceedingly difficult (*Wiggins, 1996*). In some cases *C. harwoodi* larvae are indistinguishable from other species within the genus (*Burington, 2011*). Based on our phylogenetic tree-based analysis the Crooked River *C. harwoodi* sequence groups with *C. harwoodi* sequences from Ontario (JF434099, JF434097), New Brunswick (KR146677), and Manitoba (HM102631); and not with any of the known species of *Cheumatopsyche* in British Columbia (Fig. 2). The Crooked River specimen also aligns 100% with a DNA sequence of *C. harwoodi* from Alberta (HM102632), but also with a *C. gracilis* sequence from Wyoming (HQ560573) (Fig. 2).

To identify a species based on DNA sequence, an accurate morphological identification to species of a physical specimen is required—and ideally replicated a number of times. Currently BOLD has 178 barcodes for specimens identified as *C. harwoodi* and the Crooked River specimen aligns very closely to these with less than 0.6% difference within the species as a whole, well below the 2% threshold suggested by Zhou and co-workers in 2009. There are currently only two barcodes for *C. gracilis* and both these barcodes group with the various *C. harwoodi* sequences. These two *C. gracilis* specimens are also quite different, with a 1.3% difference based on our analysis. The preponderance of evidence, then, points to one of three possibilities. First, the two *C. gracilis* specimens in BOLD are actually misidentified *C. harwoodi* and our specimen is also *C. harwoodi*. Second, the specimens represent different species but that difference is not reflected in the DNA barcode. And third, the taxonomic status of both species should be reconsidered as potentially being one species. A more definitive identification might be possible as BOLD is populated with more *C. gracilis* sequences that helps delineate the two species.

Erasmus et al. (2018), *PeerJ*, DOI 10.7717/peerj.4221

**Table 1 Trichoptera collected along the Crooked River, British Columbia and associated COI DNA barcode-assigned identifications along with date ranges of collection.** Locations of collection sites are given in the footnotes. All sequence data are available in public repositories as listed, and all specimens are vouchered at the University of Guelph—Centre for Biodiversity Genomics.

| Family[a] | Genus[a] | Species[a] | Sample IDs[b] | BIN | NCBI accession[c] | Collection site(s)[d] | Collection date range[e] | Notes |
|---|---|---|---|---|---|---|---|---|
| Brachycentridae | Brachycentrus | americanus | BIOUG18684-B11 and 22 others | BOLD:ABX6535 | KX144627 | CR2, CR2B, CR4, CR108 | 11-JUN to 13-AUG | |
| | | occidentalis | BIOUG18683-H05 and 5 others | BOLD:AAE0281 | KX144012 | CR3, CR100BR | 04-JUN to 13-AUG | |
| | Micrasema | bactro | BIOUG18683-F09.1 | BOLD:AAC4650 | KX143689 | CR4 | 11-JUN | |
| | | sp. | BIOUG18683-F08 | BOLD:ACC4912 | KX142261 | CR2 | 18-JUN | Potential new BC record |
| Hydropsychidae | Arctopsyche | grandis | BIOUG18683-A11.1 and 6 others | BOLD:AAB3049 | KX143192 | CR2, CR108 | 09-JUL to 13-AUG | |
| | Cheumatopsyche | analis | BIOUG18684-B10 | BOLD:AAA5695 | KX144608 | CR100BR | 28-JUL | |
| | | harwoodi | BIOUG18684-B09 | BOLD:AAA2316 | KX141182 | CR4 | 16-MAY | New BC record |
| | | sp. | BIOUG18684-E05 | BOLD:ACE5262 | KX142965 | CR108 | 09-JUL | |
| | | sp. | BIOUG18684-E08 and 4 others | BOLD:AAA3891 | KX142829 | CR3 | 29-JUL to 13-AUG | |
| | Hydropsyche | alhedra | BIOUG18683-H03 and 2 others | BOLD:AAC1650 | KX143172 | CR4, CR108 | 04-JUN to 11-JUN | |
| | | alternans | BIOUG18683-C12 and 14 others | BOLD:AAA3236 | KX140968 | CR3, CR100BR | 10-JUN to 13-AUG | |
| | | cockerelli | BIOUG18683-A03 | BOLD:AAC3057 | KX143078 | CR4 | 16-MAY | |
| | | morosa | BIOUG18684-E01 and 5 others | BOLD:AAA3679 | KX143491 | CR3 | 28-JUL | |
| | | slossonae | BIOUG18684-E06 and 12 others | BOLD:AAA2527 | KX143429 | CR2, CR4, CR100BR, CR108 | 11-JUN to 13-AUG | |
| Hydroptilidae | Hydroptila | arctia | BIOUG18683-F10.1 | BOLD:AAE5200 | KX141605 | CR108 | 25-JUN | |
| | | sp. | BIOUG18683-A06 | BOLD:AAK3416 | KX142062 | CR2 | 18-JUN | Potential new BC record |

Erasmus et al. (2018), *PeerJ*, DOI 10.7717/peerj.4221

**Table 1** (*continued*)

| Family[a] | Genus[a] | Species[a] | Sample IDs[b] | BIN | NCBI accession[c] | Collection site(s)[d] | Collection date range[e] | Notes |
|---|---|---|---|---|---|---|---|---|
| Lepidostomatidae | *Lepidostoma* | *pluviale* | BIOUG18684-D07.1 and 3 others | BOLD:ACF2295 | KX142857 | CR100BR | 18-JUN to 13-AUG | |
| | | sp. | BIOUG18683-G10 | BOLD:ACL5324 | KX144650 | CR2 | 4-AUG | Potential new BC record |
| | | *togatum* | BIOUG18684-D02 | BOLD:AAA2325 | KX144002 | CR3 | 14-JUL | New BC record |
| | | *cinereum* | BIOUG18683-C07.1 and 3 others | BOLD:AAK7943 | KX142572 | CR2, CR2B, CR4 | 25-JUN to 4-AUG | |
| | | *unicolor* | BIOUG18684-H04 and 8 others | BOLD:AAC5923 | KX142875 | CR4, CR108 | 11-JUN to 4-AUG | |
| Leptoceridae | *Ceraclea* | *alagma* | BIOUG18683-F06 and two others | BOLD:AAA5876 | KX143301 | CR6, CR100BR, CR108 | 16-MAY to 14-JUL | |
| | | *annulicornis* | BIOUG18683-B02 | BOLD:AAA5429 | KX142035 | CR3 | 13-AUG | New BC record |
| | | *cancellata* | BIOUG18684-A01 | BOLD:ABZ0710 | KX143326 | CR4 | 4-AUG | |
| | | *nigronervosa* | BIOUG18683-H09 and 1 other | BOLD:AAC3781 | KX141154 | CR100BR | 10-JUN | |
| | | *resurgens* | BIOUG18683-F07.1 and 2 others | BOLD:ACG9704 | KX142221 | CR3 | 14-JUL to 28-JUL | |
| Limnephilidae | *Amphicosmoecus* | *canax* | BIOUG18683-D09 and 5 others | BOLD:AAE2491 | KX143314 | CR2B, CR4, CR100BR | 11-JUN to 9-JUL | |
| | *Clistoronia* | *magnifica* | BIOUG18683-F05 and 1 other | BOLD:AAC1848 | KX141495 | CR3, CR4 | 28-JUL to 13-AUG | |
| | *Dicosmoecus* | *atripes* | BIOUG18683-G05 and 2 others | BOLD:AAC5045 | KX140940 | CR4 | 11-JUN | |
| | | *gilvipes* | BIOUG18684-H07 and six others | BOLD:AAI9526 | KX142636 | CR2B, CR4, CR100BR | 16-MAY to 9-JUL | |
| | *Limnephilus* | *externus* | BIOUG18683-F12 and 1 other | BOLD:AAA2803 | KX141731 | CR2B, CR6 | 11-JUN to 18-JUN | |
| | *Onocosmoecus* | *unicolor* | BIOUG18684-H04 and 8 others | BOLD:AAC5923 | KX142875 | CR4, CR108 | 11-JUN to 4-AUG | |
| | *Psychoglypha* | *alascensis* | BIOUG18683-G07 and 7 others | BOLD:ACH0278 | KX141905 | CR4, CR5 | 9-MAY to 4-AUG | |
| | | *subborealis* | BIOUG18683-D11.1 and 2 others | BOLD:AAE0945 | KX144814 | CR4 | 9-JUL to 4-AUG | |

**Table 1** (*continued*)

| Family[a] | Genus[a] | Species[a] | Sample IDs[b] | BIN | NCBI accession[c] | Collection site(s)[d] | Collection date range[e] | Notes |
|---|---|---|---|---|---|---|---|---|
| Philopotamidae | *Wormaldia* | *gabriella* | BIOUG18684-C03 and 4 others | BOLD:AAC1539 | KX143731 | CR2, CR108 | 21-JUL to 13-AUG | |
| Phryganeidae | *Agrypnia* | *improba* | BIOUG18683-C01 | BOLD:ACK0044 | KX143489 | CR2 | 13-AUG | |
| Polycentropodidae | *Neureclipsis* | *bimaculata* | BIOUG18683-A08 and 3 others | BOLD:AAE2683 | KX141945 | CR3 | 14-JUL to 28-JUL | |
| | *Plectrocnemia* | *cinerea* | BIOUG18684-A08 | BOLD:AAA3441 | KX141515 | CR6 | 14-JUL | |
| Rhyacophilidae | *Rhyacophila* | *brunnea* | BIOUG18683-B12 and 11 others | BOLD:AAB3088 | KX141430 | CR4, CR100BR, CR108 | 18-JUN to 2-AUG | |
| | | sp. | BIOUG18684-A07 and 3 others | BOLD:ACL4744 | KX140935 | CR2, CR100BR | 13-AUG | |
| Uenoidae | *Neophylax* | *rickeri* | BIOUG18683-G08 | BOLD:AAG9543 | KX144032 | CR4 | 4-JUN | |

**Notes.**

[a] Determined from morphological keys and BOLD database match.

[b] If more than one specimen, longest sequence from BOLD with an NCBI accession number; other sample data are available at BOLD dataset CRTRI.

[c] For the sample specified in the fourth column.

[d] CR2—54.484°N, −122.721°W; CR2B—54.484°N, −122.721°W; CR3—54.643°N, −122.743°W; CR4—54.388°N, −122.633°W; CR5—54.478°N, −122.719°W; CR6—54.328°N, −122.669°W; CR100BR—54.446°N, −122.653°W; CR108—54.458°N, −122.722°W

[e] First collection date and (if applicable) last collection date in 2014.

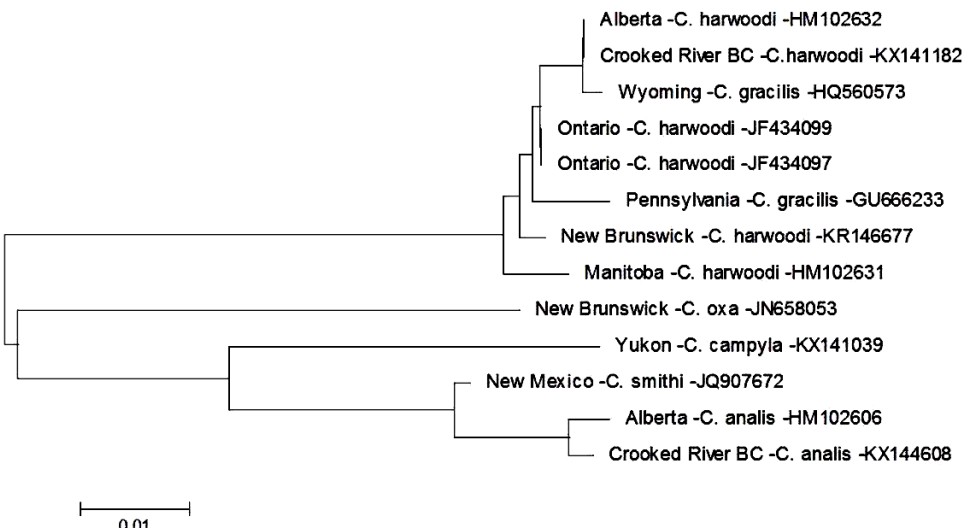

**Figure 2** **Phylogenetic tree of *Cheumatopsyche* spp. collected from the Crooked River and congeneric COI-5P DNA sequences of *Cheumatopsyche* species with DNA barcodes.** Evolutionary history is based on the Neighbour-Joining Method bootstrapped (5,000 replicates) and the Kimura-2 method to calculate distances. Each species is identified by the geographic region of collection, species, and Genbank accession number for the COI-5P DNA sequence.

On 14 July 2014 we found a larva for *Lepidostoma togatum* {synonyms *L. canadense* (Banks, 1899), *L. pallidum* (Banks, 1897), *Mormomyia togatum* Hagen, 1861, *Pristosilo canadensis* Banks, 1899, *Silo pallidus* Banks, 1897} at CR3. The DNA sequence of this specimen aligns clearly with *L. togatum* sequences (Fig. 3). Based on museum and database records in Canada *L. togatum* is known to be present in the Northwest Territories, Alberta and the Maritime Provinces of Canada. Our report is the first for this species west of the Rocky Mountains.

On 13 August 2014 we found a specimen of *Ceraclea annulicornis* {(synonyms: *Athripsodes annulicornis* (Stephens, 1836), *C. futilis* (Banks, 1914), *C. recurvata* (Banks, 1908), *Leptocerus annulicornis* Stephens, 1836, *L. futilis* (Banks, 1914)} at CR3 (Fig. 1). The phylogenetic tree-based analysis using sequences from Manitoba, Ontario, and New Brunswick strongly suggest our specimen is *C. annulicornis* (Fig. 4).

We found specimens belonging to three genera that had no significant matches at the species level on either the Barcode of Life Database or at NCBI; therefore we only provide genus-level identifications (Table 1). A specimen we putatively assign as *Micrasema* had only one match in BOLD: Genbank accession number KR145307 (*Zhou et al., 2016*), but much further south, on southern Vancouver Island. Images of this specimen are publicly available at BOLD (BIOUG18683-F08).

A specimen putatively belonging to the genus *Hydroptila* had a number of 100% matches to the Crooked River *Hydroptila* sp. in the BOLD database (*Zhou et al., 2016*), but none identified to species. Sequence alignments revealed 86% and 84.74% similarity to *H. rono* and *H. xera* respectively; both species are known to be present in British Columbia. The other two known *Hydroptila* spp. in British Columbia, *H. arctia* and *H. consimilis,* are

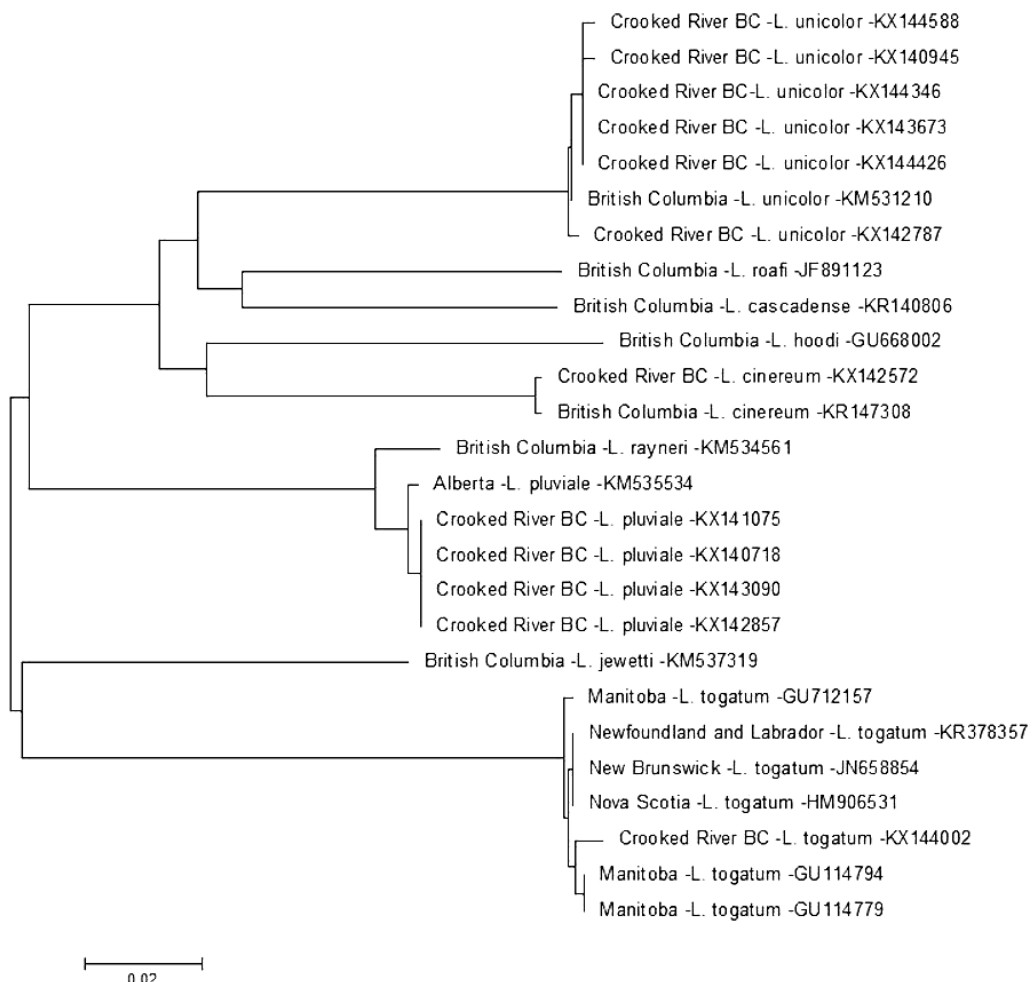

**Figure 3** **Phylogenetic tree of *Lepidostoma* spp. collected from the Crooked River and congeneric COI-5P DNA sequences of *Lepidostoma* species with DNA barcodes.** Evolutionary history is based on the Neighbour-Joining Method bootstrapped (5,000 replicates) and the Kimura-2 method to calculate distances. Each species is identified by the geographic region of collection, species, and Genbank accession number for the COI-5P DNA sequence.

substantially dissimilar from our specimen (81% and 82% match, respectively). Images of our specimen are publicly available at BOLD (BIOUG18683-A06).

A third specimen putatively assigned to *Lepidostoma* resides in a BIN with only two members (BOLD:ACL5324)—the Crooked River specimen and one other from British Columbia (Genbank Accession # KX142483). Images of this specimen (adult) are publicly available at BOLD (BIOUG18683-G10).

These three specimens are thus most likely also new species records for British Columbia. All known species in British Columbia belonging to *Micrasema* and *Hydroptila* have DNA barcodes in BOLD, and ten of the 12 *Lepidostoma* species known to be in British Columbia have DNA barcodes in BOLD. Only *L. quercina* and *L. stigma* do not, and it is possible that our specimen belongs to one of these two species.

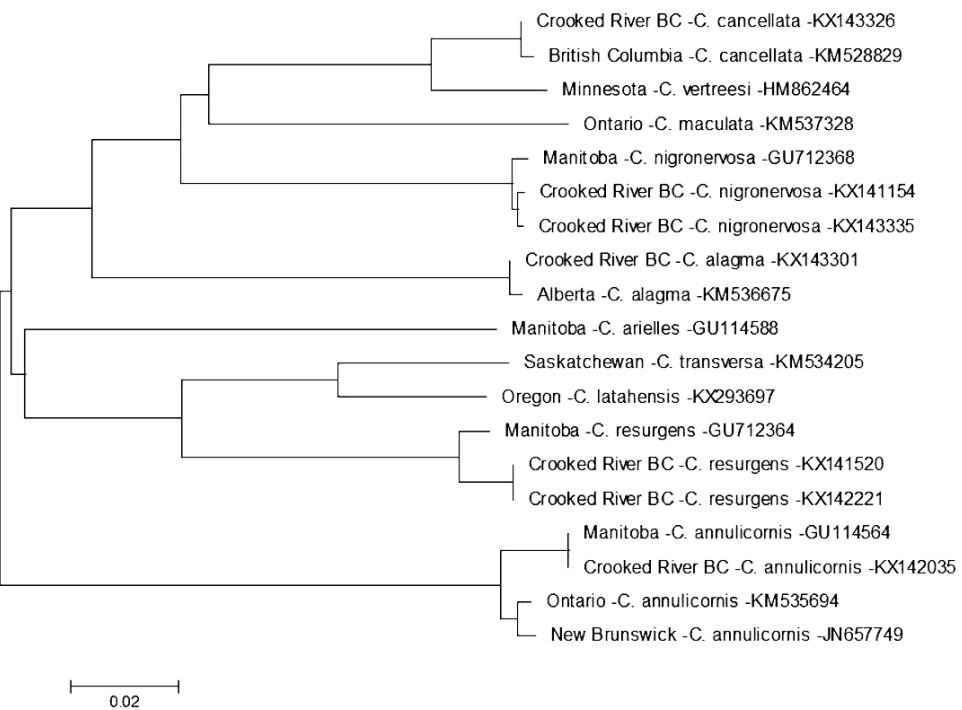

**Figure 4** **Phylogenetic tree of *Ceraclea* spp. collected from the Crooked River and congeneric COI-5P DNA sequences of *Ceraclea* species with DNA barcodes.** Evolutionary history is based on the Neighbour-Joining Method bootstrapped (5,000 replicates) and the Kimura-2 method to calculate distances. Each species is identified by the geographic region of collection, species, and Genbank accession number for the COI-5P DNA sequence.

The presence of 41 species (20 genera, 11 families) of caddisflies in the Crooked River is comparable to other rivers and regions. For instance collection from the Churchill, Manitoba area—including the Churchill River, tundra ponds, lakes, and small streams—revealed 68 species (*Zhou et al., 2009*). Collection from the Ochre River, Manitoba revealed 33 species (8 families, 17 genera) (*Cobb & Flannagan, 1990*). Broad-scale sampling across northern Canada from the Ogilvie Mountains in the Yukon to Goose Bay in Newfoundland revealed 56 species (*Cordero, Sánchez-Ramírez & Currie, 2017*). To our knowledge, there is no study that provides a comprehensive species checklist of caddisflies for a specific tributary in British Columbia to which we could compare our data more regionally.

In summary, our assessment of the Trichoptera inhabiting the Crooked River revealed three new species records for British Columbia *Lepidostoma togatum*, *Ceraclea annulicornis* and possibly *Cheumatopsyche harwoodi*. Our results also suggest at least two, and possibly three, other new species records. This baseline biodiversity data is vital for ongoing monitoring and management of this unique and highly impacted system and provides new data for managers and conservationists working in this understudied region.

## ACKNOWLEDGEMENTS

We thank Claire Shrimpton for assistance in the field. Museum databases were provided by the Beaty Biodiversity Museum at the University of British Columbia (Karen Needham and Chris Ratzlaff), the Royal British Columbia Museum (Claudia Copley and Joel Gibson), the Strickland Museum at the University of Alberta (Bryan Brunet and Felix Sperling), and the Royal Ontario Museum (Doug Currie, Antonia Guidotti, Brad Hubley, and Brenna Wells). Thank you to the reviewers, especially Dr. Ralph Holzenthal, for their feedback and comments as it improved the manuscript immensely.

### Funding

This research was funded by the University of Northern British Columbia, the Canada Research Chairs Program, the Royal British Columbia Museum, and the Canada Foundation for Innovation. The funders had no role in study design, data collection and analysis, decision to publish, or preparation of the manuscript.

### Grant Disclosures

The following grant information was disclosed by the authors:
University of Northern British Columbia.
Canada Research Chairs Program.
Royal British Columbia Museum.
Canada Foundation for Innovation.

### Competing Interests

Dezene P.W. Huber is an Academic Editor for PeerJ.

### Author Contributions

- Daniel J. Erasmus and Dezene P.W. Huber conceived and designed the experiments, performed the experiments, analyzed the data, contributed reagents/materials/analysis tools, wrote the paper, prepared figures and/or tables, reviewed drafts of the paper.
- Emily A. Yurkowski performed the experiments, analyzed the data, wrote the paper, prepared figures and/or tables.

### Field Study Permissions

The following information was supplied relating to field study approvals (i.e., approving body and any reference numbers):

Collections were completed under the British Columbia Ministry of Environment Park Use Permit #107171 where required.

### Data Availability

The raw data have been uploaded as a Supplemental File and are permanently available at 10.5883/DS-CRTRI.
## Supplemental Information

Supplemental information for this article can be found online at http://dx.doi.org/10.7717/peerj.4221#supplemental-information.

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
