# Peer review of "DNA barcode-based survey of Trichoptera in the Crooked River reveals three new species records for British Columbia"

_PeerJ, doi:10.7717/peerj.4221_

## Round 0.1 · original submission · Major Revisions

Although both reviewers recognized that your work is an important contribution, they raised a number of issues (particularly reviewer 1) that I think could potentially improve your study considerably. However, I'm optimistic that you'd be able to address them properly.

Reviewer 1 ·

Basic reporting

Introduction
Main Issues.

This work could be an important baseline for future monitoring and impact assessment. However, the way is written is more like a report and not a scientific article. There is not a clear question or hypothesis presented to be evaluated. There has been growing literature on this topic presenting great information of species list of Ephemeroptera, Trichoptera, and Plecoptera in Northern Canada and this study can be used to compare the species richness with other regions in Canada (look for Yukon and Northwest Territories EPT records in Cordero et al. 2016). Based on previous species distributions studies: how are your contributions relevant? Are the new records you present unexpected? What is the novelty: altitude, latitude? Put your study in a scientific context and not only in a report format. As well, more support in your introduction is necessary, there are many concepts or facts that are not cited

minor issues:

line 29: what portion of the COI? be concise, refer to the subunit I. Also this information needs a citation.

line 42: Support this statement. How do you know this area is true Arctic? mind that in Northern Canada, latitude alone is not an indicator of a geographic area. Recent studies have addressed this regional classification.

Experimental design

Main Issue:

The paper does not follow the format of a scientific article but looks more like a report. The research question is not stated and hence no gap in the knowledge is addressed. The contribution is not relevant the way it is presented. It needs revision about the goal of the article.

specific issues: Methods.

What is the importance of using both morphological and molecular identification analysis? Could you solve any problem presented by morphological analysis? Did you apply a sampling protocol in the different sites?

Validity of the findings

Main Issues:
The use of BOLD or other database is very useful as an initial reference. However, assumptions or conclusion made from direct comparison or the match percentage is not very strong evidence. There are many mistakes in the registered species names in the databases and the only way to confirm your findings is to put your individuals in a phylogenetic tree and see real differences. Your initial idea is good but the way how you support your findings is not enough, even to claim for first records, much less for claiming new species records. Be careful with conclusions based on arbitrary 2%

line 140: If there are no other studies how you found only three new records if all the other records should be new too? You need to dig deeper in the literature since there are good records from the 1980's for places like Yukon that are far more Isolated, I can't imagine a good amount of literature for aquatic insects in British Columbia, considering the number of studies made on altitudinal gradients in the Rocky Mountains.

Additional comments

I think the authors should be encouraged to re-think this paper. I always find very valuable information about diversity but this information should be put in a scientific context to increase the relevance of the paper as well as the journal quality. I would recommend to accept the manuscript if they address two main issues:
1. Question or hypothesis.
2. Comparison of data by the use of a phylogenetic analysis.

·

Basic reporting

This paper makes a very good contribution to the knowledge of the caddisfly fauna of an interesting river in an interesting ecosystem. The methods of analysis are relatively new to trichopterogy and represent one of the first attempts to use established DNA databases as a means of identifying local caddisfly biodiversity. However, I think the paper needs better justification as to why this method is better than the “traditional” method of collecting adult caddisflies with light traps or by netting or other means. Trichoptera species taxonomy is based on male genitalia (almost all holotypes described in the 20th century and now are males). Larvae are secondarily associated with males through rearing, use of pharate adults, or by COI sequence match. The authors should explain why they collected larvae only, or at least mention the value of adults in caddisfly taxonomy. They may have valid reasons - adult taxonomy is specialized, adults do not fly to light in cool weather, adults may fly in from other watersheds, etc. If the objective was to show that collecting larvae only was sufficient to assess the biodiversity of a stream, they should state that. It would be interesting to compare the results of a strictly larval survey with adult collecting only although I realize that this is beyond the scope of the paper. However, as presented the objectives of the paper are not expressed in context to standard caddisfly survey methods nor is it indicated their relative value.

In addition, the three new records are not placed in any context of the known distribution of the species. For example, C. harwoodi is largely an eastern North American species, so its presence in BC is surprising. This should be at least mentioned, if not discussed. Without knowing otherwise, one might conclude that the vouchers of these sequences in BOLD are misidentified.

I found only a few typos grammatical issues in the paper. I am still under the assumption that data are plural, but in some places it is used as a singular noun. Also, the authors use dashes quite a bit to separate clauses. Perhaps it is more appropriate to use commas or semicolons.

I did not find the larval photographs to be informative, although I think photographs of vouchers are required for submission to BOLD. Do these photographs match known, published descriptions of these species>

I especially find the photographs of the adult Lepidostoma to be uninformative. I can only tell that it is a Lepidostoma and it appears to be a female. There is no indication of how this specimen was obtained.

Experimental design

see above

Validity of the findings

see above

Additional comments

see above

---

## Round 0.2 · Minor Revisions

Although both reviewers were happy with the revised version of the manuscript, they identified a few minor issues that need to be addressed before the manuscript can be accepted.

Reviewer 1 ·

Basic reporting

1. The initial paragraphs of the introduction lead the reader to understand the importance of determining the biodiversity in an impacted area. However, when describing the studies area, I consider that the information from line 53 until line 74 could be summarized in a couple of lines. This information, although interesting in general, lead to ambiguity. I would recommend condensing this information.

2. In line 78, The objective number 2, states that they want to explore the biodiversity of Trichoptera. However, biodiversity has several components (alpha, beta, gamma diversity, abundance indices, ecological diversity, functional diversity, etc) that were not analyzed in this study. They only analyzed species richness and that was stated in objective 1. The objective 3 is redundant since that is expected in objective 1. I would recommend being concise with the aims and the reach of this study.

3. the sentence on line 148-150 is very confusing. The idea between DNA and morphological species identification is not clear. Please clarify this sentence.

Experimental design

The corrections suggested were done and the methods look more clear and concise. There are objectives, they were developed and the results were totally within their framework.

Validity of the findings

This study as a baseline will be very important for future references, especially in isolated areas that are impacted by human activities.

The data collected and analyzed provide support to his conclusions and can be used as a reference in future studies.

Additional comments

The corrections made by the authors were key to give clarity and relevance to this study. There are minor corrections that should be done and I mentioned above, but I recommend this paper for publication.

·

Basic reporting

Ok

Experimental design

OK

Validity of the findings

OK, although I still think the record of C. hardwoodi should be presented with some degree of uncertainty.

Additional comments

Several typographical and other errors that should be corrected and the whole paper proofread very carefully. Note comment on use of parentheses for species authors' names. Ensure bibliography is properly formatted to journal style.

---

## Round 0.3 · accepted · Accept

I'm happy with how you addressed the suggestions by the reviewers.